# A Wheeler–DeWitt Non-Commutative Quantum Approach to the Branch-Cut Gravity

Benno Bodmann [1], Dimiter Hadjimichef [2], Peter Otto Hess [3,4,*], José de Freitas Pacheco [5], Fridolin Weber [6,7], Moisés Razeira [8], Gervásio Annes Degrazia [1], Marcelo Marzola [2] and César A. Zen Vasconcellos [2,9]

[1] Departamento de Física, Universidade Federal de Santa Maria (UFSM), Santa Maria 97105-900, Brazil; benno.bodmann@acad.ufsm.br (B.B.); degrazia@ccne.ufsm.br (G.A.D.)
[2] Instituto de Física, Universidade Federal do Rio Grande do Sul (UFRGS), Porto Alegre 90010-150, Brazil; dimiter@if.ufrgs.br (D.H.); mnmarzola@gmail.com (M.M.); cesaraugustozenvasconcellos@gmail.com (C.A.Z.V.)
[3] Departamento Estructura de la Materia, Instituto de Ciencias Nucleares, Universidad Nacional Autónoma de Mexico (UNAM), México City 04510, Mexico
[4] Frankfurt Institute for Advanced Studies (FIAS), 60438 Hessen, Germany
[5] Observatoire de la Côte d'Azur, 06300 Nice, France; jose.pacheco@oca.eu
[6] Department of Physics, San Diego State University (SDSU), 5500 Campanile Drive, San Diego, CA 92182, USA; fweber@physics.ucsd.edu
[7] Center for Astrophysics and Space Sciences, University of California at San Diego (UCSD), La Jolla, CA 92093, USA
[8] Laboratório de Geociências Espaciais e Astrofísica (LaGEA), Universidade Federal do Pampa (UNIPAMPA), Caçapava do Sul 96570-000, Brazil; moisesrazeira@unipampa.edu.br
[9] International Center for Relativistic Astrophysics Network (ICRANet), 65122 Pescara, Italy
* Correspondence: hess@nucleares.unam.mx; Tel.: +52-55-56233386

**Abstract:** In this contribution, motivated by the quest to understand cosmic acceleration, based on the theory of Hořava–Lifshitz and on the branch-cut gravitation, we investigate the effects of non-commutativity of a mini-superspace of variables obeying the Poisson algebra on the structure of the branch-cut scale factor and on the acceleration of the Universe. We follow the guiding lines of a previous approach, which we complement to allow a symmetrical treatment of the Poisson algebraic variables and eliminate ambiguities in the ordering of quantum operators. On this line of investigation, we propose a phase-space transformation that generates a super-Hamiltonian, expressed in terms of new variables, which describes the behavior of a Wheeler–DeWitt wave function of the Universe within a non-commutative algebraic quantum gravity formulation. The formal structure of the super-Hamiltonian allows us to identify one of the new variables with a modified branch-cut quantum scale factor, which incorporates, as a result of the imposed variable transformations, in an underlying way, elements of the non-commutative algebra. Due to its structural character, this algebraic structure allows the identification of the other variable as the dual quantum counterpart of the modified branch-cut scale factor, with both quantities scanning reciprocal spaces. Using the iterative Range–Kutta–Fehlberg numerical analysis for solving differential equations, without resorting to computational approximations, we obtained numerical solutions, with the boundary conditions of the wave function of the Universe based on the Bekenstein criterion, which provides an upper limit for entropy. Our results indicate the acceleration of the early Universe in the context of the non-commutative branch-cut gravity formulation. These results have implications when confronted with information theory; so to accommodate gravitational effects close to the Planck scale, a formulation à la Heisenberg's Generalized Uncertainty Principle in Quantum Mechanics involving the energy and entropy of the primordial Universe is proposed.

**Keywords:** branch-cut cosmology; Wheeler–DeWitt equation; non-commutative quantum gravity

## 1. Introduction

As an ontological domain-extended version of General Relativity [1], analytically continued to the complex plane and combined with the multiverse conception of Hawking–Hertog [2], the branch-cut gravitation (BCG) [3–10] is the result of the axiomatic incorporation of the mathematical principles and norms of existential closure and completeness [11]. Domain extension in Quantum Mechanics, through the incorporation of complex variables [12], not only broadened our perception of the submicroscopic world, but also revealed direct physical manifestations associated with infinitesimal small scales [13,14]. At the other extreme, assuming an environment encompassed by pseudo-complex General Relativity (pc-GR), such a descriptive notion of domain extension brought to light a suppression mechanism of the primordial gravitational singularity and the prediction of the existence of dark energy outside and inside cosmic mass distributions [15–17], with unique consequences for the stability of compact stars and for the evolution of the Universe.

In this contribution, we focus our study on BCG since, by covering General Relativity in a extended domain to the complex plane, this formulation represents one of the most promising theories to describe the early evolutionary stages of the Universe.

A serious problem in the quantization of General Relativity (GR) is that this theory, as demonstrated by a simple power counting, is not renormalizable. Fortunately, the Hořava–Lifshitz [18,19] formulation is a renormalizable theory of gravity that is also Lorentz invariant at low energies, although it breaks this symmetry at high energies, which is a consequence of an implicitly present minimum length. In the early Universe, the idea of a minimal length becomes important and has to be addressed. A minimal length can be included in various manners and is related not only to a breaking of Lorentz invariance but also to the appearance of a non-commutative behavior in short space-time scales. Due to the effects of quantum gravity, the effective space-time dimensionality can change in the UV regime, implying that point sources are effectively smoothed by the Planck scale characteristics of non-commutative quantum fields [20].

In this contribution, we address the effects of the non-commutativity of a mini-superspace of variables obeying the Poisson algebra on the structure of the branch-cut scale factor and on the acceleration of the Universe by means of a non-commutative algebraic formulation of the Hořava–Lifshitz theory. In Section 2, we review the concepts of the classical BCG approach, and in Section 3, the basic motivation for a non-commutative quantum gravity is addressed. In Section 4, the commutative BCG approach is outlined, and in Section 5, the extension to the non-commutative approach is discussed in detail. In the same section, the Wheeler–DeWitt equation is rewritten in a convenient form, enabling numerical to be found solutions for the wave function of the Universe. In Section 6, new numerical solutions are presented, and the results of the non-commutative BCG quantum gravity are compared to its commutative version. In Section 7, a discussion of the results is presented, and, finally, in Section 8, conclusions are drawn.

## 2. Classical BCG Approach

The classical version of BCG [3–10] describes a foliated Universe in which multiple singularities merge, generating a topological smooth branch-cut structure of Riemann surfaces, continuously connected, with a new scale parameter, $\ln^{-1}[\beta(t)]$, analytically continued to the complex plane, the only dynamical variable of the theory.[1] The branched manifold $\mathcal{M}$ is layered on hypersurfaces, $\Sigma_t$, restricted to Riemann foliation leaves, characterized by a complex time parameter, $t$, with the analytically continued branching line element defined as [4,5]

$$ds^2_{[\text{ac}]} = -\sigma^2 N^2(t)c^2 dt^2 + \sigma^2 \left(\ln^{-1}[\beta(t)]\right)^2 \left[\frac{dr^2}{(1-kr^2(t))} + r^2(t)\left(d\theta^2 + sin^2\theta d\phi^2\right)\right]. \quad (1)$$

In this expression, the variables $r$ and $t$, respectively, represent real and complex space-time parameters, while $k$ represents the spatial curvature of the multiverse, corresponding to

negatively curved ($k = -1$), flat ($k = 0$), or positively curved ($k = 1$) spatial hypersurfaces. $N(t)$ in turn represents the lapse function[2] with $\sigma^2 = 2/3\pi$ denoting a normalization factor.

BCG theory additionally contemplates analytically continued Friedmann-type equations, as well as expressions for the energy–stress conservation law, Hubble rate, deceleration parameter, Ricci scalar, Ricci curvature, and the corresponding complex conjugated expressions (for details, see [3–10]). The classical version of the BCG theory describes a smooth Universe with a fine-tuned transition region circumscribing the contraction and the expansion phases, purely geometric in nature, that replaces the cosmological singularity.

Therefore, the primordial singularity is replaced by a family of Riemann foliation sheets. These sheets depict the branching cosmic scale factor $\ln^{-1}[\beta(t)]$, which when shrinking to a finite critical size, is shaped by a range, foliation regularization, and domain extension encoded by the $\beta(t)$ function. The range of this function extends beyond the Planck dimensions according to Bekenstein's criterion.[3]

Bekenstein's criterion establishes an upper limit on the thermodynamic entropy contained in a given finite region of space, as well as the corresponding (finite) amount of associated energy. Alternatively, the criterion establishes the maximum amount of information needed to perfectly describe a given physical system, down to the quantum level. The Bekenstein bound states that the entropy of a given region of space-time—where gravity is so strong that nothing, including light or other electromagnetic waves, has enough energy to escape it—is proportional to the number of Planck areas that would be needed to cover the corresponding event horizon. In this sense, there is perfect harmony between the Planck dimensions and the dimensions of the event horizon. In this domain, translating this view to the corresponding BCG predictions [7], which indicates a significantly larger range of the branched cosmic scale factor in comparison to the Planck dimensions, this range can be interpreted as quantifying the number of Planck areas that would be necessary to cover the primordial singularity, thus reconciling the BCG predictions with the micro-structure of (quantum) space-time.

In the contracting phase of the branch-cut Universe, as each patch size decreases with a linear dependence on $\ln^{-1}[\beta(t)]$, light travels through geodesics in each Riemann foliation sheet, continuously bordering each branch point, and although the horizon size scales with $\ln^{\epsilon}[\beta(t)]$, where $\epsilon$ denotes the dimensionless thermodynamic connection, the length of the path for the light to travel compensates for the difference in scale between the patch and horizon sizes. Under these conditions, causality between the horizon size and the patch size can be achieved through the accumulation of branches in the transition region between the current state of the Universe and the primordial past of the events [7]. In addition to causality, the flatness and horizon problems of cosmology have been addressed by BCG theory [7,10]. The flatness problem involves a very small Planck value for the scaling ratio between the total density of the Universe and the critical density, $\Omega_c \sim \ln^{2\epsilon}[\beta(t)] / \ln^2[\beta(t)]$ [23–25]. The horizon problem, in turn, is related to the lack of a causal connection between the patch size of the observable Universe and the past of events [23–25].

As a corollary, the branched gravitation approach merely expands the realm of realization of the governing principles of General Relativity, while maintaining the rationality, structural logics, mathematical operations, and norms underlying its theoretical foundations. BCG's classic and quantum formulations, in tune with its inspiring theoretical motivations, seek to shed new light and fresh perspectives on the question of the origin and evolution of the Universe. In the following, we recall recently developed BCG quantum approaches, within the scope of commutative quantum gravity, and subsequently, we introduce elements of the present formulation, which seeks to advance the field of a non-commutative theoretical proposal.

## 3. Non-Commutative Quantum Gravity

The intense inhomogeneous and anisotropic nature of the primordial expansion generated causally disconnected patches leading to the assumption of a period of cosmic inflation

during which the exponential growth of the Universe contributed to the smoothing of heterogeneities within the cosmological horizon, defining, in this way, the observable causal domain.

The ΛCDM model establishes that our Universe is presently in a phase of accelerated expansion originating in the presence of dark energy, while cold dark matter (CDM) is the main driver of the gravitational interaction, responsible in turn for shaping the large-scale structure of the Universe. This view of the evolutionary Universe has been supported during the last few decades by means of a series of main cosmological probes, for instance, the cosmic microwave background (CMB) [26,27], type Ia supernovae [28–30], and baryonic acoustic oscillations [31,32].

To the best of our knowledge, H.S. Snyder was the first to propose a quantized space-time [33]. In the abstract of his paper, he notes that "it is usually assumed that space-time is a continuum. This assumption is not required by Lorentz invariance. In this paper we give an example of a Lorentz-invariant discrete space-time". The emphasis is that space-time can be quantized without violating Lorentz symmetry. This publication was mostly ignored. In [34], it is shown that a pseudo-complex extension of coordinates and momenta leads to a non-commutative quantization of the real coordinates and momenta. Lorentz invariance is also conserved. This formulation has, as a consequence, a minimal length, which simulates the appearance of a quantized space-time. Numerous other attempts have been made to quantize space-time, with string theory and quantum-loop quantization being the most prominent. However, these efforts generally suffer from the drawback of violating Lorentz invariance, resulting in complex calculations. Whether or not one should prioritize the preservation of Lorentz invariance depends on the specific context under consideration.

In this manuscript, the Hořava–Lifshitz theory is used, which keeps Lorentz invariance at low energy but violates it at high energy, which corresponds to large momenta and thus to very small distances. In this context, we demonstrate the possibility of introducing a non-commutative formulation, enabling the study of the effects of a minimal length on the Universe's solutions. As just discussed, it is important to study non-commutativity of space-time because, at high energy (small distances), the structure of space-time must undergo changes to yield finite results in quantum theory, particularly in General Relativity, which is our interest here.

In recent years, the quest to understand the acceleration of the Universe has led to numerous propositions and scenarios, including non-commutative quantum cosmologies. In what follows, we investigate the effects of non-commutativity of mini-superspace variables on the accelerating behavior of the branch-cut cosmic scale factor and the wave function of the Universe.

## 4. Commutative Quantum BCG Approach

Recently, we proposed Lagrangian formulations for the BCG in [9,10] within the framework of Hořava–Lifshitz's theory of renormalizable gravity (HLGT) [18,19]. This approach contemplates high-order curvature terms while preserving General Relativity diffeomorphism [35] and the usual foliation of the Arnowitt–Deser–Misner (ADM) formalism in the infrared limit [36]. In combination with the Wheeler–DeWitt (WdW) equation [37], the formulation is free of ghosts, making it suitable for describing quantum effects of the gravitational field [36]. The solutions of the WdW equation, represented in turn by a geometric functional of compact manifolds and matter fields, describe the evolution of the quantum wave function of the Universe [38,39]. The most intriguing aspect of the WdW equation, the absence of the time variable, although linked to a classical image of space-time [35], represents a feature of the classical Hamilton–Jacobi formulation of General Relativity, in which the observable universe does not exhibit time-reversal symmetry, giving way to a quantum description of events, as the basic cosmic constituents [40–44]. The wave function, in turn, depends only on a "3-geometry", which corresponds to the equivalence class of metrics under a diffeomorphism. It does not rely on the specific coordinate-dependent form of the metric tensor [42–44]. In terms of cosmological quantum interpretation, the

wave function of the Universe can be expressed as a functional restricted to a superspace configuration, which includes three-surface and matter fields denoted by $\Phi$. The metric is represented by $h_{ij}$ in this framework. The associated WdW wave function, denoted as $\Psi(h_{ij}, \Phi)$, can be understood as describing the evolution of $\Psi(\Phi)$ regarding the physical variable $\Phi$.

The action of the projectable Hořava–Lifshitz theory combined with BCG, $\mathcal{S}_{HL}$, depends on the branching scalar curvature of the Universe, $\mathcal{R}$, and its derivatives, in different orders [9,10,18,19]:

$$
\begin{aligned}
\mathcal{S}_{HL} = \ & \frac{M_P^2}{2} \int d^3x \, dt \, N\sqrt{g} \left( K_{ij}K^{ij} - \lambda K^2 - g_0 M_p^2 - g_1 \mathcal{R} - g_2 M_P^{-2} \mathcal{R}^2 \right. \\
& - g_3 M_P^{-2} \mathcal{R}_{ij} \mathcal{R}^{ij} - g_4 M_P^{-4} \mathcal{R}^3 - g_5 M_P^{-4} \mathcal{R}(\mathcal{R}_j^i \mathcal{R}_i^j) - g_6 M_P^{-4} \mathcal{R}_j^i \mathcal{R}_k^j \mathcal{R}_i^k \\
& \left. - g_7 M_P^{-4} \mathcal{R} \nabla^2 \mathcal{R} - g_8 M_P^{-4} \nabla_i \mathcal{R}_{jk} \nabla^i R^{jk} \right),
\end{aligned}
\tag{2}
$$

where $\lambda$ and $g_i$ represent running coupling constants[4], $M_P$ is the Planck mass, $\nabla_i$ denotes covariant derivatives, and the branching Ricci components of the three-dimensional metrics are determined by imposing a maximum symmetric surface foliation [9]:

$$
\mathcal{R}_{ij} = \frac{2}{\sigma^2 \ln^{-2}[\beta(t)]} g_{ij}, \quad \text{and} \quad \mathcal{R} = \frac{6}{\sigma^2 \ln^{-2}[\beta(t)]}.
\tag{3}
$$

In expression (2), $K$ represents the trace of the extrinsic curvature tensor $K_{ij}$, given by [9,10]

$$
K = K^{ij} g_{ij} = -\frac{3}{2\sigma N} \frac{\left(\frac{d}{dt} \ln^{-1}[\beta(t)]\right)}{\ln^{-1}[\beta(t)]}.
\tag{4}
$$

By introducing the variable change $u(t) \equiv \ln^{-1}[\beta(t)]$, with $du \equiv d\ln^{-1}[\beta(t)]$, and subsequently applying standard canonical quantization procedures,[5] and by promoting the canonical conjugate momentum into an operator, i.e., $p_u \mapsto -i\frac{\partial}{\partial u}$, the Hamiltonian is also elevated to an operator. The new complex dynamical variable $u(t)$, representing the helix-like scale factor analytically continued to the complex plane, along with the corresponding conjugate momentum $p_u$, are then treated as operators, denoted, respectively, as $\hat{\mathcal{H}}(t)$, $\hat{u}(t)$, and $\hat{p}_u$. This leads to the formulation of the branching Hamiltonian given by[6] [19]

$$
\mathcal{H} = \frac{1}{2} \frac{N}{u(t)} \left[ -p_u^2 - g_k u^2(t) + g_\Lambda u^4(t) + g_r + \frac{g_s}{u^2(t)} \right], \quad \text{with} \quad p_u = -\frac{u(t)}{N} \frac{du(t)}{dt}.
\tag{5}
$$

In this expression, $p_u$ represents the conjugate momentum of the original branching gravitation dynamical variable $\ln^{-1}[\beta(t)]$, $g_k$, $g_\Lambda$, $g_r$, and $g_s$ represent respectively the curvature, cosmological constant, radiation, and stiff matter running coupling constants [19,45]

$$
g_k \equiv \frac{2}{3\lambda - 1}; \quad g_\Lambda \equiv \frac{\Lambda M_P^{-2}}{18\pi^2(3\lambda - 1)^2}; \quad g_r = 24\pi^2(3g_2 + g_3);
$$
$$
g_s \equiv 288\pi^4(3\lambda - 1)(9g_4 + 3g_5 + g_6).
\tag{6}
$$

The $g_r$ and $g_s$ running coupling constants can be positive or negative, without affecting the stability of the solutions. The contribution of stiff matter, in turn, is determined by the condition $\rho = \omega p$ in the corresponding equation of state. In [10], we supplemented the Hamiltonian with two additional terms, $g_m u$, which describes the contribution of baryon matter combined with dark matter, and $g_q u^3$, a quintessence contribution, a time-varying,

spatially inhomogeneous and negative pressure component of the cosmic fluid [46,47], which allows approaching the "coincidence problem".[7]

## 5. Non-Commutative Quantum BCG Approach

In order to incorporate a non-commutative quantum formulation, we adhere to the principles outlined in [48,49], with a slight alteration to the coordinate transformations in the classical phase-space, resulting in a super-Hamiltonian that is more in line with the canonical structure of the commutative formulation, as we demonstrate below.

To build a formalism with non-commutative variables, the authors of [48,49] introduced in the Hořava–Lifshitz formalism the action of a perfect fluid, characterized by a dimensionless number $\omega$, associated with a dual variable to the scale factor of the standard formulation $a(t)$, represented by $v(t)$, whose canonically conjugated momentum is represented by $p_v$. After inserting this contribution into Equation (5), supplemented with the two additional terms $g_m u$ and $g_q u^3$, the following expression is obtained

$$\mathcal{H} = \frac{1}{2} \frac{N}{u(t)} \left[ -p_u^2 + g_r - g_m u - g_k u^2 - g_q u^3 + g_\Lambda u^4 + \frac{g_s}{u^2(t)} + \frac{1}{u^{3\omega-1}} p_v \right], \quad (7)$$

with $p_v = -\frac{v(t)}{N} \frac{dv(t)}{dt}$. The condition for late time acceleration, imposed on the equation of the state parameter of dark energy, corresponds to $\omega < -1/3$, where $\omega$ is the ratio of pressure $p$ and the energy density $\rho$.

In the next steps, the authors of [49], based on a similar Hamiltonian formulation, sought to determine a Lagrangian density of the system by using an iterative procedure and introducing the following non-commutative algebra:

$$\{u, v\} = \sigma; \quad \{p_u, p_v\} = \alpha; \quad \{u, p_v\} = \gamma; \quad \{v, p_u\} = \chi; \quad \{u, p_u\} = \{v, p_v\} = 1; \quad (8)$$

translating this composition to the branch-cut gravity formulation, due to the complex nature of the variable $u$, the algebraic relation (8) states that the variable $v$ as well as the set of variables $\{\sigma, \alpha, \gamma, \chi\}$ must also be complex.

Still, according to the assumptions outlined in [49], the following coordinate transformation in the classical phase-space was defined:

$$\tilde{u} = u; \quad \tilde{v} = v; \quad \tilde{p}_u = \frac{1}{\Gamma} \left( -p_u + 2\alpha v + \chi p_v \right); \quad \tilde{p}_v = \frac{1}{\Gamma} \left( \gamma p_u + \alpha u - p_v \right), \quad (9)$$

with

$$\Gamma = (\alpha\sigma - 1) + \chi\gamma \quad \rightarrow \quad \Gamma = \chi\gamma - 1, \quad (10)$$

since $\sigma = 0$ is the condition for the new variables $\tilde{u}$ and $\tilde{v}$ to satisfy the Poisson bracket algebra[8] [49]. By inverting the above Equation (9), in combination with the non-commutative Hamiltonian (7), a super-Hamiltonian then results, with the following dependence on the variables $\tilde{u} = u, \tilde{v} = v, \tilde{p}_u, \tilde{p}_v, \tilde{\partial}_u, \tilde{\partial}_v$ (for comparison, see [49]):

$$
\begin{aligned}
\mathcal{H} &= \frac{1}{2} \frac{N}{u(t)} \left[ -p_u^2 + g_r - g_m u - g_k u^2 - g_q u^3 + g_\Lambda u^4 + \frac{g_s}{u^2(t)} + \frac{1}{u^{3\omega-1}} p_v \right], \\
&= \frac{1}{2} \frac{N}{u(t)} \left[ -\left( -\tilde{p}_u + 2\alpha v + \chi \tilde{p}_v \right)^2 + \frac{1}{u^{3\alpha-1}} \left( \gamma \tilde{p}_u + \alpha u - \tilde{p}_v \right) \right. \\
&\quad \left. + \left( g_r - g_m u - g_k u^2 - g_q u^3 + g_\Lambda u^4 + \frac{g_s}{u^2(t)} \right) \right].
\end{aligned}
\quad (11)
$$

Making the replacements $\tilde{p}_u \to -i\frac{\tilde{\partial}}{\partial u}$ and $\tilde{p}_u \to -i\frac{\tilde{\partial}}{\partial v}$, Equation (11) may be rewritten as

$$
\begin{aligned}
\mathcal{H} = & \; \frac{1}{2}\frac{N}{u(t)}\left[\left(\frac{\tilde{\partial}^2}{\partial u^2} - 4i\alpha v\frac{\tilde{\partial}}{\partial u} - \chi\left[\frac{\tilde{\partial}}{\partial u}\frac{\tilde{\partial}}{\partial v} + \frac{\tilde{\partial}}{\partial v}\frac{\tilde{\partial}}{\partial u}\right] + 4i\alpha\chi v\frac{\tilde{\partial}}{\partial v} + \chi^2\frac{\tilde{\partial}^2}{\partial v^2} - 4\alpha^2 v^2\right)\right. \\
& \; \left. - \frac{1}{u^{3\alpha-1}}\left(i\gamma\frac{\tilde{\partial}}{\partial u} - i\frac{\tilde{\partial}}{\partial v} - \alpha u\right) + \left(g_r - g_m u - g_k u^2 - g_q u^3 + g_\Lambda u^4 + \frac{g_s}{u^2(t)}\right)\right].
\end{aligned}
\tag{12}
$$

As an alternative, in view of the formal structure resulting from the transformations (9), in what follows, a symmetrical treatment in terms of the variables $u$ and $v$ is proposed, and ambiguities in the ordering of the quantum operators $\frac{\tilde{\partial}}{\partial u}\frac{\tilde{\partial}}{\partial v} + \frac{\tilde{\partial}}{\partial v}\frac{\tilde{\partial}}{\partial u}$ are overcome by adopting a different phase-space coordinate transformation, which allows a consistent symmetric incorporation of the non-commutative algebra into the intrinsic structure of the new set of variables $\{\tilde{u}, \tilde{p}_u, \tilde{v}, \tilde{p}_v\}$. We then introduce the following linear mapping, which relates the commutative $\{u, p_u, v, p_v\}$ and the non-commutative phase-space set of variables $\{\tilde{u}, \tilde{p}_u, \tilde{v}, \tilde{p}_v\}$ in the super-Hamiltonian:

$$
\tilde{u} = u; \quad \tilde{v} = v; \quad \tilde{p}_u = \frac{1}{\Gamma}\left(p_u - \chi p_v\right); \quad \tilde{p}_v = \frac{1}{\Gamma}\left(-\gamma p_u + \alpha u - \alpha v + p_v\right).
\tag{13}
$$

As this is not the main topic of this contribution, for a more profound discussion about factor ordering in Quantum Mechanics and quantum gravity and its implications for the behavior of the wave function of the Universe, see [10,50–52].

From Equation (13), we obtain, by incorporating the $\Gamma$ matrices in the conjugate momenta $\tilde{p}_u$ and $\tilde{p}_v$:

$$
\tilde{p}_u = \left(p_u - \chi p_v\right); \quad \tilde{p}_v = \left(-\gamma p_u + \alpha u - \alpha v + p_v\right).
\tag{14}
$$

By inverting Equation (13),

$$
\tilde{p}_u \to p_u = \left(\tilde{p}_u - \chi\tilde{p}_v\right); \quad \tilde{p}_v \to p_v = \left(-\gamma\tilde{p}_u + \alpha u - \alpha v + \tilde{p}_v\right),
\tag{15}
$$

the super-Hamiltonian becomes

$$
\begin{aligned}
\mathcal{H} = & \; \frac{1}{2}\frac{N}{u(t)}\left[-p_u^2 + g_r - g_m u - g_k u^2 - g_q u^3 + g_\Lambda u^4 + \frac{g_s}{u^2(t)} + \frac{1}{u^{3\omega-1}}p_v\right], \\
= & \; \frac{1}{2}\frac{N}{u(t)}\left[-\left(\tilde{p}_u - \chi\tilde{p}_v\right)^2 - \frac{1}{u^{3\alpha-1}}\left(\gamma\tilde{p}_u - \alpha u + \alpha v - \tilde{p}_v\right)\right. \\
& \; \left. + \left(g_r - g_m u - g_k u^2 - g_q u^3 + g_\Lambda u^4 + \frac{g_s}{u^2(t)}\right)\right].
\end{aligned}
\tag{16}
$$

The Hamiltonian operator given in Equation (16), when applied to the wave function of the Universe, $\Psi(u, v)$, under the condition $\mathcal{H}\Psi(u, v) = 0$, gives the following equation:

$$
\begin{aligned}
\left[\left(\frac{\tilde{\partial}^2}{\partial u^2} - 2\chi\frac{\tilde{\partial}}{\partial u}\frac{\tilde{\partial}}{\partial v} + \chi^2\frac{\tilde{\partial}^2}{\partial v^2}\right) + \frac{1}{u^{3\alpha-1}}\left(i\gamma\frac{\tilde{\partial}}{\partial u} - i\frac{\tilde{\partial}}{\partial v} + \alpha u - \alpha v\right)\right. \\
\left. + \left(g_r - g_m u - g_k u^2 - g_q u^3 + g_\Lambda u^4 + \frac{g_s}{u^2(t)}\right)\right]\Psi(u, v) = 0.
\end{aligned}
\tag{17}
$$

In what follows, the parameters $\chi$ and $\gamma$ are treated as complex numbers; however, to make contact with conventional formulations, especially regarding the insertion of a set of ordering factors to overcome ambiguities in the ordering of quantum operators, and maintain the complex nature of the variables $u$ and $v$, we consider only the real component

of the parameter $\alpha$. From now on, we use the following notation for formal simplification: $\gamma = i|\gamma|$, so $i\gamma = i^2|\gamma| = i\gamma = -|\gamma|$, so the previous equation becomes:

$$\left[\left(\frac{\tilde{\partial}^2}{\partial u^2} - 2\chi\frac{\tilde{\partial}}{\partial u}\frac{\tilde{\partial}}{\partial v} + \chi^2\frac{\tilde{\partial}^2}{\partial v^2}\right) - \frac{1}{u^{3\alpha-1}}\left(|\gamma|\frac{\tilde{\partial}}{\partial u} + i\frac{\tilde{\partial}}{\partial v} - \alpha u + \alpha v\right)\right.$$
$$\left. + \left(g_r - g_m u - g_k u^2 - g_q u^3 + g_\Lambda u^4 + \frac{g_s}{u^2(t)}\right)\right]\Psi(u, v) = 0. \tag{18}$$

Additionally, for simplicity, we do not use the symbol and $\tilde{\ }$ in the partial derivatives as identification of the new variables in the scope of non-commutative algebra. Equation (18) can thus be rewritten in the general form:

$$\left[a(u, v)\frac{\partial^2}{\partial u^2} - 2b(u, v)\frac{\partial^2}{\partial u\partial v} + c(u, v)\frac{\partial^2}{\partial v^2}\right]\Psi(u, v)$$
$$= \left[d(u, v)\frac{\partial}{\partial u} + e(u, v)\frac{\partial}{\partial v} + F(u, v)\right]\Psi(u, v),$$
$$= G\left(u, v, \Psi, \frac{\partial}{\partial u}, \frac{\partial}{\partial v}\right), \tag{19}$$

with

$$a(u, v) = 1; \quad b(u, v) = \chi; \quad c(u, v) = \chi^2; \quad d(u, v) = \frac{\gamma}{u^{3\alpha-1}}; \quad e(u, v) = \frac{i}{u^{3\alpha-1}};$$
$$F(u, v) = -\left(g_r - g_m u - g_k u^2 - g_q u^3 + g_\Lambda u^4 + \frac{g_s}{u^2} + \frac{\alpha}{u^{3\alpha-2}} - \frac{\alpha v}{u^{3\alpha-1}}\right), \tag{20}$$

where $a(u, v)$, $b(u, v)$, and $c(u, v)$ represent functions of the independent variables $u$ and $v$, and have continuous derivatives up to the second-order. Since the $b^2(u, v) - a(u, v)c(u, v) = 0$ expression (19) belongs to the mathematical group of parabolic equations, to reduce this equation to a canonical form, one should first write out the characteristic equation [53]

$$a\,dv^2 - 2b\,du\,dv + c\,du^2 = 0, \tag{21}$$

which splits into two equations

$$a\,dy - \left(b \pm \sqrt{b^2 - ac}\right)dx = 0. \tag{22}$$

Then, one should find their general integrals. In the case of a parabolic equation, the two previous solutions coincide, resulting in a common general integral $\varphi(u, v) = \mathcal{I}_G$. This allows the variables $u$ and $v$ to be changed to new independent variables $\xi$ and $\eta$, in accordance with

$$\xi = \varphi(u, v), \quad \text{and} \quad \eta = \eta(u, v), \tag{23}$$

where $\eta(u, v)$ is a differentiable function that satisfies the non-degeneracy condition of the Jacobian $D(\xi, \eta)/D(x, y)$ in the given domain. As a result, Equation (19) is reduced to the canonical form

$$\frac{\partial^2 \Psi(\xi, \eta)}{\partial \eta^2} = G\left(\xi, \eta, \Psi, \frac{\partial}{\partial \xi}, \frac{\partial}{\partial \eta}\right). \tag{24}$$

For $\eta$ one can take $u$ or $v$. We take, for convenience, $u$. In the Faddeev–Jackiw formalism, the variables $u$ and $v$ are non-commutative, and after the variable transformation, $\xi$ and $\eta$ are also non-commutative. In this sense, $\xi$ and $\eta$ are canonically conjugate dual vari-

ables, which span reciprocal spaces, so the following relation between these variables holds:

$$\xi = \frac{1}{\sqrt{2\pi}} \int_{-\infty}^{\infty} A(\eta) e^{i\xi\eta} d\eta \,. \tag{25}$$

Equation (17) then becomes

$$
\begin{aligned}
\frac{\partial^2 \Psi(\xi,\eta)}{\partial \eta^2} &= \left( d(\xi,\eta)\frac{\partial}{\partial \eta} + e(\xi,\eta)\frac{\partial}{\partial \xi} + F(\xi,\eta) \right)\Psi(\xi,\eta)\,, \\
&= -\left( \frac{\gamma}{\eta^{3\alpha-1}}\frac{\partial}{\partial \eta} + g_r - g_m\eta - g_k\eta^2 - g_q\eta^3 + g_\Lambda\eta^4 + \frac{g_s}{\eta^2} + \frac{\alpha}{\eta^{3\alpha-2}} \right)\Psi(\xi,\eta) \\
&\quad + \left( \frac{i}{\eta^{3\alpha-1}}\frac{\partial}{\partial \xi} - \frac{\alpha\xi}{\eta^{3\alpha-1}} \right)\Psi(\xi,\eta)\,.
\end{aligned}
\tag{26}
$$

This equation may be rearranged in the form

$$\mathcal{H}(\xi,\eta)\Psi(\xi,\eta) = 0\,, \tag{27}$$

from which the super-Hamiltonian $\mathcal{H}(\xi,\eta)$ may be identified as:

$$
\begin{aligned}
\mathcal{H}(\xi,\eta) = \frac{1}{2}\frac{N}{\eta(t)}\Bigg[ &\frac{\partial^2}{\partial \eta^2} + \frac{\gamma}{\eta^{3\alpha-1}}\frac{\partial}{\partial \eta} + g_r - g_m\eta - g_k\eta^2 - g_q\eta^3 + g_\Lambda\eta^4 + \frac{g_s}{\eta^2} \\
&+ \frac{\alpha}{\eta^{3\alpha-2}} - \frac{\alpha\xi}{\eta^{3\alpha-1}} - \frac{i}{\eta^{3\alpha-1}}\frac{\partial}{\partial \xi} \Bigg]\,.
\end{aligned}
\tag{28}
$$

For comparison with the original version of the super-Hamiltonian (7), we may rewrite Equation (28) in the form

$$
\begin{aligned}
\mathcal{H}(\xi,\eta) = \frac{1}{2}\frac{N}{\eta(t)}\Bigg[ &-p_{\eta,\gamma,\alpha}^2 + g_r - g_m\eta - g_k\eta^2 - g_q\eta^3 + g_\Lambda\eta^4 + \frac{g_s}{\eta^2} \\
&+ \frac{\alpha}{\eta^{3\alpha-2}} - \frac{\alpha\xi}{\eta^{3\alpha-1}} + \frac{1}{\eta^{3\alpha-1}}p_\xi \Bigg]\,,
\end{aligned}
\tag{29}
$$

with

$$-p_{\eta,\gamma,\alpha}^2 \equiv \frac{\partial^2}{\partial \eta^2} + \frac{\gamma}{\eta^{3\alpha-1}}\frac{\partial}{\partial \eta}\,. \tag{30}$$

The super-Hamiltonian given in expression (29) shows that the adoption of the particular phase-space transformation in the context of Poisson algebra given by expression (13) brings additional advantages. One advantage refers, as we will see below, to the maintenance of the original formal structure, providing consistent comparisons between the predictions of the commutative and non-commutative formulations. The commutative Hořava–Lifshitz formulation presented in Equation (7), which served as a starting point for the alternative non-commutative algebraic formulation (13), following similar procedures performed in [49], contemplates a standard formulation that contains the quadratic term $\hat{p}_{\eta,\gamma,\alpha}^2$, where $\hat{p}_{\eta,\gamma,\alpha}$ represents the operator corresponding to the canonical moment associated with the variable $\eta$, and the contribution corresponding to its dual variable $\xi$ is represented by the linear term $\hat{p}_\xi$. The transformations introduced in (13), despite originally also containing second-order derivatives referring to both variables $u$ and $v$, including the contribution originally referring to the perfect fluid, through the canonical reduction of the original differential equation, by means of a transformation of coordinates delineated by the corresponding characteristic equations, allow one to identify and select one of the new coordinates $\xi$ or $\eta$ in a very particular way, restricting, in this way, the

resulting Hamiltonian to a dependence on the canonical momenta $\hat{p}_{\eta,\gamma,\alpha}^2$ and $\hat{p}_\xi$, in a similar way to the original mathematical structure of the commutative algebraic formulation. This is because, although one of the original variables can be directly identified with one of the new transformed coordinates, the other is restricted to identification with the common general integral. In short, carrying out an adequate identification, the new variables, although underlying their dual character, allow a transcription that follows, from a formal point of view, the main structure of the commutative formulation, thus allowing a consistent comparison between the evolutionary processes of the wave function of the Universe, considering the commutative and the non-commutative algebraic formulations. Another advantage concerns the form of the expression (30) for the quadratic term referring to the variable $\eta$, which includes a term dependent on a first-order partial derivative $\partial/\partial\eta$, multiplied by a factor given by $\gamma/\eta^{3\alpha-1}$. The form of this expression is structurally similar to the insertion of a set of ordering factors in quantum gravity formulations, given by $\kappa$, in order to overcome ambiguities in ordering quantum operators. A way of approaching this theme can be exemplified in the expression below (see, for example, [10])

$$p_\kappa^2 = -\frac{1}{\eta^\kappa(t)}\frac{\partial}{\partial\eta(t)}\left(\eta^\kappa(t)\frac{\partial}{\partial\eta(t)}\right). \tag{31}$$

Taking as an example $\kappa = 0, 1, 2$, for comparison with standard formulations (see [10] and references therein), we obtain, from (31), respectively

$$p_0^2 = -\frac{\partial^2}{\partial\eta(t)^2}; \quad p_1^2 = -\left\{\frac{\partial^2}{\partial\eta(t)^2} + \frac{1}{\eta}\frac{\partial}{\partial\eta(t)}\right\}; \quad p_2^2 = -\left\{\frac{\partial^2}{\partial\eta(t)^2} + \frac{2}{\eta}\frac{\partial}{\partial\eta(t)}\right\}. \tag{32}$$

Thus, in the order of the removal of ambiguities in the ordering of quantum operators, the choices $\kappa = 0, 1, 2$ in combination with (30) would lead to the following set of values for the $\gamma$ and $\alpha$ parameters [9]: for $p_0$, $\gamma = \alpha = 0$; for $p_1$, $\gamma = 1, \alpha = 2/3$; and for $p_2$, $\gamma = 2; \alpha = 2/3$. In what follows, we consider that the parameters $\gamma$ and $\alpha$ are normalized to 1, in which the negative values of these parameters are not excluded. The numerical approach of the authors of [49], in calculating the dependence of the scale factors as a function of the parameter $\alpha$, considering $\alpha = -0.5, 0, 05$, reveals that such variations imply very small changes in the scale factors. This gives us confidence in choosing the value $\alpha = 1/3$, which makes it possible to separate the variables in the super-Hamiltonian equation for the wave function of the Universe, in order to solve the corresponding partial differential equation, despite its known technical difficulties, without approximations. Finally, the formal structure of the super-Hamiltonian obtained allows us to identify the new variable $\eta(t)$ with the branch-cut gravitation scale factor, $\ln^{-1}[\beta(t)]$, although evidently incorporating, as a result of the imposed variables' transformations, in an underlying way, elements that characterize a non-commutative algebra. Due to its structural character, this algebraic structure allows identifying the complex variable $\xi(t)$ as the complex dual quantum counterpart of $\eta(t)$, both scanning reciprocal quantum complex spaces.

### 5.1. Friedmann-Type Branch-Cut Equations

In view of the previous interpretation, the non-commutative BCG contemplates complex equations similar to Friedmann's equations (for comparison see for instance [10]):

$$\left(\frac{\frac{d}{dt}\eta(t)}{\eta(t)}\right)^2 = \frac{8\pi G}{3}\rho(t) - \frac{kc^2}{\eta(t)} + \frac{1}{3}\Lambda; \quad \left(\frac{\frac{d^2}{dt^2}\eta(t)}{\eta(t)}\right) = -\frac{4\pi G}{3}\left(\rho(t) + \frac{3}{c^2}p(t)\right) + \frac{1}{3}\Lambda, \tag{33}$$

where $\Lambda$ represents the cosmological constant, as well as the corresponding complex conjugated expressions:

$$\left(\frac{\frac{d}{dt}\eta^*(t^*)}{\eta^*(t^*)}\right)^2 = \frac{8\pi G}{3}\rho^*(t^*) - \frac{kc^2}{\eta^*(t^*)} + \frac{1}{3}\Lambda^*; \quad \left(\frac{\frac{d^2}{dt^2}\eta^*(t^*)}{\eta^*(t^*)}\right) = -\frac{4\pi G}{3}\left(\rho^*(t^*) + \frac{3}{c^2}p^*(t^*)\right) + \frac{1}{3}\Lambda^*. \quad (34)$$

These are the equations that underlie the scenarios of the non-commutative branched gravitation in the imaginary sector, in which the primordial singularity is replaced by a foliated transition region, described by the helix-like cosmological factor $\eta(t)$, analytically continued to the complex plane, interposing two distinct evolutionary stages of the Universe, a contraction and an expansion phase. The consequences of these scenarios on the behavior of the wave function of the universe are notable insofar as they imply the evolutionary description of $\Psi(\eta)$ in both regions.

### 5.2. Boundary Conditions

Approaches to quantum gravity based on the Hořava–Lifshitz formulation usually found in the literature, due to the technical computational difficulties, are limited frequently to plotting potentials and/or to solving the corresponding Friedmann and Wheeler–DeWitt equations by using approximation methods, and, in the context of non-commutative algebra, those approaches impose restrictions into the parameter space, in order to simplify the formal treatment in the search of viable solutions. In this contribution, we do not use numerical computational approximations, although we have limited the parameter $\alpha$ to the value $1/3$ to allow for the separation of variables in the super-Hamiltonian associated with the wave function of the Universe. The parameter $\chi$ in turn, as already stressed, in view of the nature of the variable transformations, is implicitly contained in the definition of $\zeta$, which together with the quantity $\eta$, form a fundamental dual system and two complementary elements of the theory, the rescaled cosmic scale factor $\eta$ and its counterpart $\zeta$, defined in a reciprocal space.

The differential equations presented in this work were solved adhering to established principles and standard criteria to ensure convergence, stability, and continuity of the solutions. Using the iterative Range–Kutta–Fehlberg numerical analysis for solving differential equations, we obtained numerical solutions, with the boundary conditions of the wave function of the Universe based on the Bekenstein criterion, which provides an upper limit for the entropy, following the proposition presented in [10].

The entropy of a black hole, according to the Bekenstein limit, is proportional to the number of Planck areas needed to cover the black hole's event horizon (where each Planck area corresponds to one entropy unit). In non-commutative branched gravitation, we assume that the primordial singularity is equally covered by a certain number of Planck areas, the number value of which in turn corresponds to the primordial entropy of the Universe. Assuming that the dimensions of this border region correspond to the most distant points that can be casually observed, taking the proper distance $d(t)$ of a pair of objects, at any arbitrary instant $t$ and its relation to the proper distance $d(t_0)$ at a reference time $t_0$, such that $d(t) = |\eta(t)|d(t_0)$, this implies that for $t = t_0$, then $|\eta(t_0) = 1|$. From a quantum probabilistic point of view, this condition implies a maximum probability of observation, $|\Psi(1)| = 1$, assuming a normalized wave function. Thus, the boundary conditions assumed in this contribution in the contraction sector are $\Psi(-1) = -1$, and in the expansion region are $\Psi(1) = 1$.

### 6. Solutions for the Wave Function of the Universe

With the particular choice $\alpha = 1/3$, which allows a separation of variables, Equation (26) reduces to

$$
\begin{aligned}
\frac{\partial^2 \Psi(\xi, \eta)}{\partial \eta^2} &= -\left( \gamma \frac{\partial}{\partial \eta} + g_r + \frac{1}{3}\eta - g_m\eta - g_k\eta^2 - g_q\eta^3 + g_\Lambda\eta^4 + \frac{g_s}{\eta^2} \right) \Psi(\xi, \eta) \\
&+ \left( i\frac{\partial}{\partial \xi} - \frac{1}{3}\xi \right) \Psi(\xi, \eta).
\end{aligned}
\tag{35}
$$

Assuming expression (35) is separable, adopting the representation $\Psi(\xi, \eta) = \Psi(\xi)\Psi(\eta)$, the following equations then hold:

$$
\left( \frac{\partial^2}{\partial \eta^2} + \gamma\frac{\partial}{\partial \eta} + g_r + \frac{1}{3}\eta - g_m\eta - g_k\eta^2 - g_q\eta^3 + g_\Lambda\eta^4 + \frac{g_s}{\eta^2} - \mathcal{C} \right) \Psi(\eta) = 0;
$$

$$
\rightarrow \left( \frac{\partial^2}{\partial \eta^2} + \gamma\frac{\partial}{\partial \eta} + \tilde{g}_r - \tilde{g}_m\eta - g_k\eta^2 - g_q\eta^3 + g_\Lambda\eta^4 + \frac{g_s}{\eta^2} \right) \Psi(\eta) = 0;
$$

$$
\rightarrow \left( \frac{\partial^2}{\partial \eta^2} + \gamma\frac{\partial}{\partial \eta} + V(\eta) \right) \Psi(\eta) = 0,
\tag{36}
$$

with $\tilde{g}_r \equiv g_r - \mathcal{C}$, $\tilde{g}_m \equiv g_m - 1/3$,

$$
V(\eta) = \tilde{g}_r - \tilde{g}_m\eta - g_k\eta^2 - g_q\eta^3 + g_\Lambda\eta^4 + \frac{g_s}{\eta^2},
\tag{37}
$$

and

$$
\left( i\frac{\partial}{\partial \xi} - \frac{1}{3}\xi + \mathcal{C} \right) \Psi(\xi) = 0,
\tag{38}
$$

where $\mathcal{C}$ is a constant. The solution to the second equation above, up to an additional constant, is

$$
\Psi(\xi) = i\left( \mathcal{C}\xi - \frac{1}{3}\xi^2 \right).
\tag{39}
$$

In order to make contact with previous calculations, we adopt values for the running coupling constants from [9,10]).

Figures 1 and 2 (below) show the behavior of the effective potential for different sets of values of the running coupling constants corresponding to the non-commutative formulation. The results show a domain of the term that describes stiff matter. The formulation adopted contemplates, in a still preliminary stage of the theoretical approach, real potentials, although the wave function $\Psi(\eta)$ and the scale factor $\eta$ are complex quantities. Therefore, in this preliminary approach, we did not adopt a formulation that would allow, as with the classic BCG formulation, to circumvent the singularity, which we intend to address in a subsequent investigation. Fortunately, the boundary conditions imply a topological leap in the region where singularities predominate, allowing a formulation consistent with the implications of the Bekenstein limit. The main difference between the plots in Figures 1 and 2 is related to the factor $g_s$. In Figure 1, $g_s$ has a negative value, generating an attractive potential that contemplates a singularity, while in Figure 2, $g_s$ is positive, generating a barrier of potential with two disconnected regions. The asymptotic behavior of the curves, however, does not present significant differences for the adopted values of the running coupling constants.

Considering that the change in sign of the parameters $\tilde{g}_r$ and $g_s$ does not affect the stability of the solutions, Figure 3 shows the behavior of the potential $V(\eta)$ with a change in the sign of the running coupling constant of the dominant term $g_s/\eta^2$, which represents

the stiff matter contribution. The implications of these results in terms of signatures that discriminate these parameterizations, however, require a more rigorous future analysis. According to the curves in Figure 3, the potential (37) shows a similar behavior both in the commutative and in the non-commutative case, being practically insensitive to the new Poisson algebra parameters introduced.

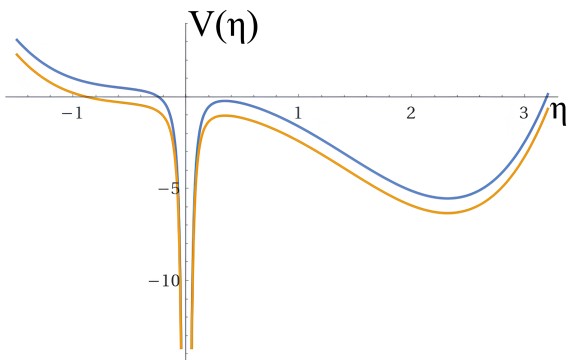

**Figure 1.** Graphical illustration of the potential $V(\eta)$ (Equation (37)) corresponding to the non-commutative approach. The values for the running coupling constants are: $\tilde{g}_r = 0.4$; $\tilde{g}_m = 0.6185$; $g_k = 1$; $g_q = 0.7$; $g_\Lambda = 0.333$; $g_s = -0.03$ (blue line), and $\tilde{g}_r = -0.4$; $\tilde{g}_m = 0.6185$; $g_k = 1$; $g_q = 0.7$; $g_\Lambda = 0.333$; $g_s = -0.03$ (yellow line).

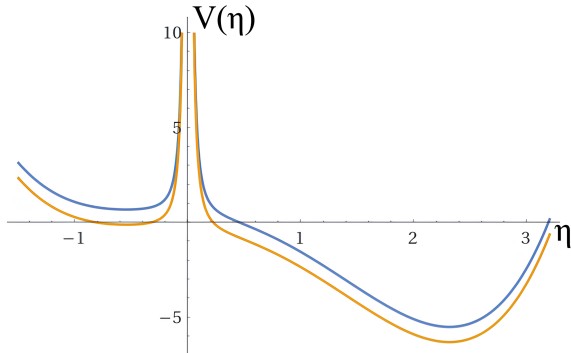

**Figure 2.** Same as Figure 1 but for values of the running coupling constants given by $\tilde{g}_r = 0.4$; $\tilde{g}_m = 0.6185$; $g_k = 1$; $g_q = 0.7$; $g_\Lambda = 0.333$; $g_s = 0.03$ (blue line), and $\tilde{g}_r = -0.4$; $\tilde{g}_m = 0.6185$; $g_k = 1$; $g_q = 0.7$; $g_\Lambda = 0.333$; $g_s = 0.03$ (yellow line).

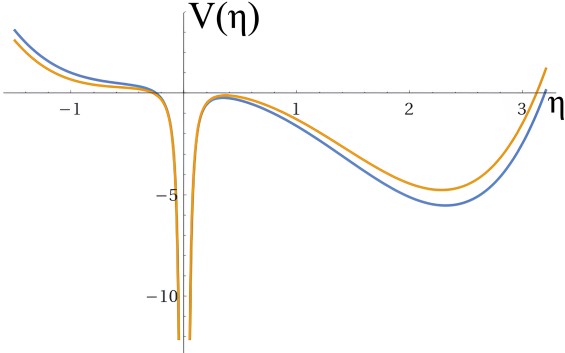

**Figure 3.** Graphical illustration of the potential $V(\eta)$ for the commutative (yellow line) and non-commutative (blue line) approaches. The values of the running coupling constants are: $\tilde{g}_r = 0.4$; $\tilde{g}_m = 0.2855$; $g_k = 1$; $g_q = 0.7$; $g_\Lambda = 0.333$; $g_s = -0.03$ (yellow line), and $\tilde{g}_r = 0.4$; $\tilde{g}_m = 0.6185$; $g_k = 1$; $g_q = 0.7$; $g_\Lambda = 0.333$; $g_s = -0.03$ (blue line).

Figure 4 presents typical solutions of the wave function of the Universe corresponding to the commutative case. The wave function of the Universe, as a function of the branch-cut scale factor, presents a wave-like behavior whose amplitudes progressively increase, in the contraction sector, when approaching the boundary region, and shows the opposite behavior in the expansion sector, where the amplitudes of the wave function of the Universe decrease as the wave function moves away from the boundary region. These results describe the evolution of the wave function of the Universe—associated with hypersurfaces $\Sigma_\eta$ analytically continued to the complex plane—in the cosmic scale factor $\eta(t)$. The main characteristics of these solutions are the oscillatory behavior, whose increasing amplitudes during the Universe contraction are contrasted by the decreasing of the thermodynamic entropy, while, in the expansion sector, the corresponding main characteristics are the decreasing amplitudes of the wave function and increasing of the thermodynamic entropy. These results imply a Universe described by oscillating quantum states tending towards a stable ordering at some future time. Going back to the past of the events, the systematic increase in the oscillatory amplitudes of the wave function as a function of the scale factor $\eta$ suggests the accumulation of branches, as indicated by the BCG, to restore causality. The effect of accumulating branches actually occurs in both phases corresponding to the expansion and contraction regions around the transition domain modulated by an accumulation of Riemann sheets. The oscillatory behavior of the wave function of the Universe corresponding to the commutative case, in the contraction sector, implies a period of acceleration, prior to the branch-cut transition. Likewise, the corresponding oscillatory behavior in the expansion phase could be interpreted as a deceleration of the branch-cut Universe, in disagreement with the inflation canons. Conversely, the results of the non-commutative model indicate an inflationary period, a natural consequence of the non-commutative branch-cut gravity structure, without the need to introduce this assumption in an ad hoc way, as we see below.

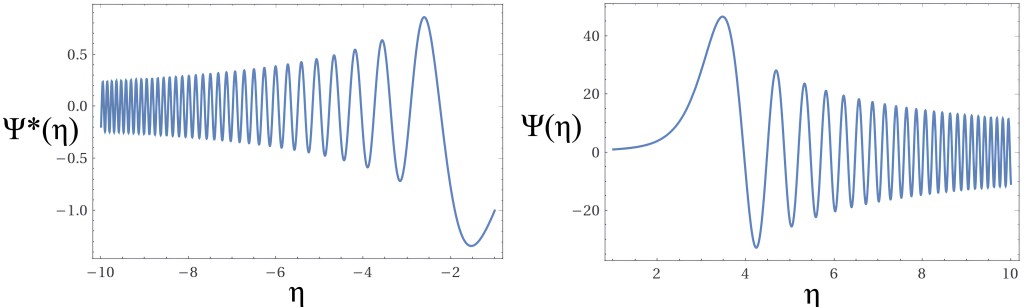

**Figure 4.** Typical solutions of the wave function $\Psi(\eta)$ corresponding to the commutative algebra, with $\alpha = \chi = \gamma = 0$, $g_r = 0.4$, and $g_s = -0.03$.

Figure 5 presents results of the wave function of the Universe corresponding to the non-commutative case with $\gamma = -1$ and $\alpha = 1/3$. The results indicate that the wave function of the Universe, as a function of the branch-cut scale factor, in the contraction and expansion sectors, presents a similar wave-like behavior, growing systematically in both phases. This behavior indicates, as a result of the imposition of the non-commutativity of a mini space-time superspace of variables that obey the Poisson algebra, acceleration of the branched Universe. In the contraction sector, the behavior of the wave function of the Universe is similar to the behavior in the corresponding commutative sector; the most striking difference is thus registered in the expansion sector, indicating an inflationary period of our Universe. This behavior is repeated for other choices of running coupling constants, except for the case in which we change the sign of the parameter $\gamma$, as we see below. We limit the number of curves displayed in order not to overcrowd graphs with results similar to this contribution, impairing the reading dynamics.

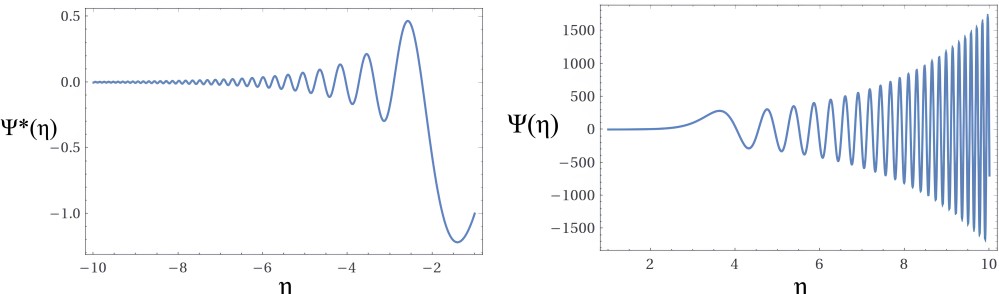

**Figure 5.** Solutions of the wave functions $\Psi^*(\eta)$ (**left**) and $\Psi(\eta)$ (**right**), corresponding to the non-commutative algebra, with $\gamma = -1$. The values of the running coupling constants are: $\tilde{g}_r = 0.4$; $\tilde{g}_m = 0.6185$; $g_k = 1$; $g_q = 0.7$; $g_\Lambda = 0.333$; $g_s = -0.03$.

Figure 6 shows the results of the wave function of the Universe corresponding to the non-commutative case with $\gamma = 1$ and $\alpha = 1/3$. The figures show that the amplitudes of the wave function of the Universe, as a function of the branch-cut scale factor, in the contraction and expansion sectors, present a wave-like behavior similar to the previous figures. However, unlike the previous results, the amplitudes systematically decrease in both phases. This behavior indicates, as a result of the imposition of the non-commutativity of a mini space-time superspace of variables that obey the Poisson algebra, deceleration of the branched Universe. In the expansion sector, the behavior of the wave function of the Universe is similar to the behavior in the corresponding commutative sector; the most striking difference is thus registered in the contraction sector, indicating a disinflationary period of the branched Universe.

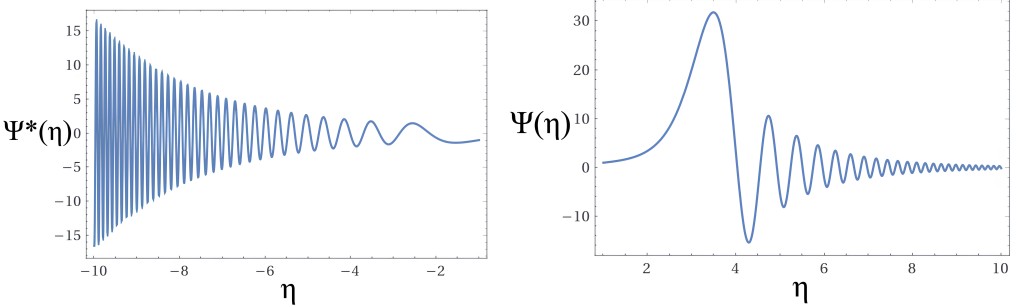

**Figure 6.** Solutions of the wave functions $\Psi^*(\eta)$ (**left**) and $\Psi(\eta)$ (**right**), corresponding to the non-commutative algebra, with $\gamma = 1$. The values of the running coupling constants are: $\tilde{g}_r = 0.4$; $\tilde{g}_m = 0.6185$; $g_k = 1$; $g_q = 0.7$; $g_\Lambda = 0.333$; $g_s = -0.03$.

## 7. Discussion of the Results

There are numerous works in the literature that deal with the topic of quantum gravity in a non-commutative environment. With regard to the Hořava–Lifshitz quantum gravity formalism combined with the Wheeler–DeWitt equation, or other cosmological models, the number of articles decreases considerably. These works deal, in most cases, with the temporal evolution of the Universe's scale factor in a standard formalism, characterized by the nomenclature designation *dynamical equations*, with the number of authors dealing with the evolution of the Universe's wave function being quite limited. In some articles, the authors deal with both themes (see, for example, for theoretical approaches [54–65] and for an experimental contribution [66]). Most authors, with regard to the wave function of the Universe and the resolution of its evolutionary equations, central themes in our investigation, direct their studies towards formal aspects, and when looking for numerical or algebraic results, due to the computational difficulties imposed by the formalism, use approximations that significantly limit their conclusions.

Although this is not the central theme of this work, it is important to highlight that studies involving strings and non-commutative gauge theories have contributed significantly to a better understanding of the influence of the non-commutative algebra on geometric structures and the accelerated evolution of our Universe. Starting with Edward Witten's brilliant work published in 1986 [67], many authors have followed this line of investigation (for a review up to 2005 see [68]). Among the most recent works, we highlight [69] (and references therein) whose cosmological solutions describe a Universe in accelerated expansion, with several realizations in string theory models.

In the present work, as previously mentioned, we used the method known as Runge–Kutta–Fehlberg for solving differential equations, without adopting any numerical approximation. Our numerical calculations enabled a broad study of the evolution of the function $\Psi(\eta)$ and a wide class of solutions, the presentation of which, for reasons of brevity, we limit to just a few figures. These results, involving the contraction phase as well as the expansion phase of the branch-cut Universe, as far as we know, are original, and indicate an acceleration of the wave function $\Psi(\eta)$ in the expansion phase, in tune with the predictions of the inflation model, as well as a deceleration in the contraction phase, with both predictions being in tune with the BCG predictions.

Our results have implications when confronted with information theory. For a given random variable $\zeta$ with $n$ possible event values (outcomes) $q_1, \ldots, q_n$ such that the probability of each outcome is denoted by $Prob[\zeta = q_i] = \mathcal{P}_i$, the information equation, defined in the form

$$I(\mathcal{P}) = -log_b(\mathcal{P}) = log_b\left(\frac{1}{\mathcal{P}}\right), \tag{40}$$

relates the degree of information associated to a particular event, represented by $I(\mathcal{P})$, and the associated probability $\mathcal{P}$ that this event may occur. The amount of information in a random message $\zeta$ is given by Shannon information entropy [70], defined as

$$H(\zeta) := \sum_{i=1}^{n} \mathcal{P}_i I_i(\mathcal{P}_i) = \sum_{i=1}^{n} \mathcal{P}_i log_b\left(\frac{1}{\mathcal{P}_i}\right). \tag{41}$$

In more precise terms, in information theory, $H(\zeta)$ represents the average number of bits needed to encode a random message. For every random variable $\zeta$ distributed on a set of $n$ values, Shannon's entropy obeys the inequality

$$0 \leq H(\zeta) \leq 1/n. \tag{42}$$

$H(\zeta) = 0$ occurs if and only if a distribution is concentrated at one point, and $H(\zeta) = 1/n$, if and only if the distribution is uniform. From the point of view of the corresponding interpretation, in information theory, a minimum entropy corresponds to the maximum probability that a certain event occurs, whereas a maximum entropy occurs when all probabilities of all outcomes have equal quantified values, more precisely, as $1/n$. This conception of information theory reinforces the idea that the entropy at the beginning of the Universe is close to zero and cannot be null since, according to the Bekenstein criterion, it would be impossible for singularities to occur, from a thermodynamic point of view. Recalling that the Bekenstein criterion imposes that the initial state of the Universe is unique, therefore, in a probabilistic conception, the primordial state of the Universe would fit the case of minimum entropy as theorized in the theory of information. In the process of the formation of a black hole, the catalyzed conversion of a pure quantum state to a mixed state occurs, in contradiction with the principle of unitary quantum evolution, thus causing loss of information. To reach a value close to zero, the thermodynamic entropy decreases in the contraction phase, consequently increasing in the expansion phase. This realization has led to the "information paradox", a topic that has been the scene of fierce conceptual disputes.

According to Quantum Mechanics, however, the information content of isolated systems is conserved. On the other hand, entropy subadditivity seems to describe information overload when examining single components of a composite system and their correlations in the case where we disregard the intrinsic quantum information encoded in the coherence of pure states. To overcome this gap, by introducing the concept of coherent entropy, necessary to account for the "missing" information, it is possible to restore its conservation [71]. Furthermore, coherent entropy is equal to the information transmitted in the future by the quantum states. These concepts, when translated into the quantum-evolutionary description of the cosmic wave function, indicate that this asymmetric behavior finds, in information theory, a safe harbor to indicate that the informational content of the branch-cut Universe does not change in time when we consider the phases of contraction and expansion in an associated integrated manner.

From the point of view of the last information transmitted from the contraction sector to the expansion sector, associated with the boundary conditions, imagining that the temporal parametric propagation takes place in the direction of the transition region, the results displayed in Figure 4 indicate that this information is preserved. Despite the last transmitted information, in view of the boundary conditions, $\Psi(-1) = -1$ and $\Psi(1) = 1$, corresponding to the quantum leap between the phases that separate the early Universe and the present Universe, which could seem to imply non-conservation of information, the values of the probability densities corresponding to the two phases of the wave function of the Universe, $|\Psi(\pm 1)| = 1$, coincide, thus implying the confirmation of this conservation.[10]

The conservation of the last information referring to the contraction and expansion phases would not be confirmed, apparently, in a preliminary view of the results of Figures 5 and 6 since the apex of the amplitudes of the wave functions corresponding to the contraction phase of the early Universe do not correspond to the starting point of the amplitudes in the expansion regions of the present Universe. This is because, due to the structure of the non-commutative differential equations, the starting point of the solution in the expansion phase in Figure 5 corresponds to a null value, while in Figure 6, the opposite result occurs. Similar results occur for positive values of $g_s$.

However, these results, in light of classical branch-cut gravitation, require further analysis. The classic BCG view, as we stated earlier, describes a smooth Universe with a fine-tuned transition region circumscribing the contraction and the expansion phases, purely geometric in nature, that replaces the cosmological singularity. The results obtained through the imposition of the Bekenstein criterion contemplate two interpretations, one from the classical point of view and the other from the quantum point of view. The classical view would indicate, as we also stated earlier, that the primordial singularity is replaced by a family of Riemann foliation sheets that depict the branched cosmic scale factor $\ln^{-1}[\beta(t)]$ shrinking to a finite critical size, shaped by the range, foliation regularization, and domain extension function $\beta(t)$. Moreover, as we saw in [7], the range of this function extends beyond the Planck dimensions, representing a gateway, from the classical point of view, of all the information contained in the evolutionary process of the Universe traversing from the contraction to the expansion phases, generating a communication channel between the primordial Universe and the present one, serving as a source of primordial seeds.

From this point of view, it would be essential to reconcile BCG's classical and quantum views and establish a minimal length consistent with the Planck dimensions, and this compatibility process would indicate that, from the point of view of information theory, Figures 5 and 6 seem to indicate, unlike what is well established by Quantum Mechanics, that information in the present case is not conserved, being generated, or lost, in the transition process between the contraction and expansion phases depending on the values, in this case, of the crucial parameter $\gamma$ in the modeling of the non-commutative branch-cut algebraic formulation. However, to deepen this issue, it is necessary to go back to the step of introducing new variables, $\eta$ and $\xi$, with a view to solving the wave equation of the Universe $\Psi(\eta, \xi)$ described in terms of the separated components $\Psi(\eta)$ and $\Psi(\xi)$, implying,

let us say, two linearly independent random variables, represented by $\zeta$ and $\varsigma$, which obey the requirements of a joint conditional information entropy

$$H(\zeta, \varsigma) := \sum_{i=1}^{n} \sum_{j=1}^{m} \mathcal{P}_{i,j} I_{i,j}(\mathcal{P}_{i,j}) = \sum_{i=1}^{n} \sum_{j=1}^{m} \mathcal{P}_{i,j} log_b \left( \frac{1}{\mathcal{P}_{i,j}} \right). \quad (43)$$

The conditional entropy is a measure of how much uncertainty remains about the random variables $\zeta$ and $\varsigma$.

With respect to the Bekenstein bound, a topic that has aroused permanent discussions in the search for compatibility between General Relativity and Quantum Gravitation, an interesting path, within the scope of the formulation of non-commutative BCG, would be to resort, as carried out, for example, by the authors of [72], to a modified/extended version of the Heisenberg uncertainty relation, in light of quantum gravity, to accommodate gravitational effects close to the Planck scale. This is a crucial problem, still open in the domain of quantum gravity.

Bekenstein's universal upper bound of a localized quantum system establishes that

$$S \leq \frac{2\pi k_B R E}{\hbar c}, \quad (44)$$

with $E$ representing the total energy of a system enclosed in a circumference with surface area $A$ and radius $R$. Assuming a particle in a nutshell with mass $m$, characterized by a wave packet of spatial size $R$ and linear momentum $p = E/c$ (see, for instance, [72]), from expression (44) the following uncertainty relation can be defined:

$$\Delta S \Delta E \lesssim \frac{2\pi k_B}{\hbar c} \Delta R (\Delta E)^2 \quad \rightarrow \quad \Delta S \Delta E \lesssim \frac{2\pi k_B}{\hbar c} \frac{(\Delta R \Delta E)^2}{\Delta R}; \quad (45)$$

where, since the motion of the particle is unknown a priori, the following assumptions $\Delta p \simeq p_x \simeq \Delta E/c$ and $\Delta x \simeq R$, as the uncertainty on its position, were adopted. From Equations (44) and (45), we obtain

$$\Delta S \Delta E \lesssim \frac{2\pi k_B}{\hbar c} \frac{(\Delta R \Delta E)^2}{\Delta R} \simeq \frac{2\pi k_B c^2}{\hbar c} \frac{(\Delta x \Delta p)^2}{\Delta x}. \quad (46)$$

This expression, in combination with the Generalized Heisenberg Uncertainty Principle (GHUP), may be recast as

$$\Delta S \Delta E \lesssim \frac{2\pi k_B c^2}{\hbar c} \frac{(\Delta x \Delta p)^2}{\Delta x} \gtrsim \frac{2\pi k_B c^2}{\hbar c} \frac{1}{\Delta x} \frac{\hbar^2}{4} \left[ 1 + \theta \left( \frac{\Delta p}{mc} \right)^2 \right] \simeq \frac{\pi k_B \hbar c}{2 \Delta x} \left[ 1 + \theta \left( \frac{\Delta p}{mc} \right)^2 \right], \quad (47)$$

where $\theta$ is a parameter that describes the scale at which quantum-gravitational effects become relevant. From this expression, the regime $\Delta p \simeq mc$ results in

$$\Delta S \Delta E \lesssim \frac{\hbar c}{2} \frac{\pi k_B (1 + \theta)}{\sqrt{\theta} \ell_P}, \quad (48)$$

in which $\sqrt{\theta}$ represents the minimum scale in the non-commutative space-time algebra, so the minimum observable GHUP length is assumed as $\Delta x \sim \sqrt{\theta} \ell_P$.

Figure 7 outlines three different perspectives for the transition region between the contraction and expansion phases of branched gravitation. The figure on the left presents a sketch of the classical view in which the Bekenstein criterion would shape the dimensions of the "transition portal". The middle figure sketches the quantum view in which the Bekenstein criterion shapes the region contemplating a quantum leap. The figure on the right shapes a conception still under construction in which both previous conceptions consistently intersect, in which Bekenstein's criterion shapes both regions. This conception

would imply the possibility of accessing distances typical of the Planck scale, and at the same time would allow seeds from the contraction phase to flow into the expansion phase. This would imply, however, that a small but finite amount of entropy or information can be packed into a region of the Planck dimensions needed to cover the primordial singularity, thus reconciling the classical view of BCG predictions with the micro-structure of space-time.

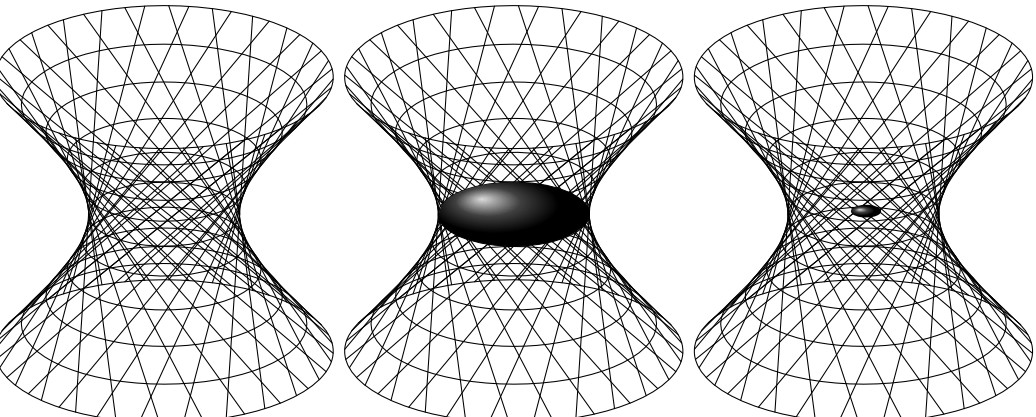

**Figure 7.** The figure outlines three different perspectives for the transition region between the contraction and expansion phases of branched gravitation: on the left is the classical view in which Bekenstein criterion would shape the dimensions of the "transition portal"; in the middle is the quantum view in which the Bekenstein criterion shapes the region contemplating a quantum leap; on the right is a conception in which both previous conceptions consistently intersect, in which Bekenstein criterion shapes both regions. Figure produced with Tikz by one of the authors (CAZV).

## 8. Conclusions

General Relativity is described in terms of a commutative space-time geometry, while Quantum Mechanics is described in terms of non-commutative algebras generated by position and momentum or energy and time operators satisfying canonical commutation relations. Understanding how to reconcile General Relativity with the concepts of Quantum Mechanics represents a challenge for a consistent quantum theory of gravity, as it involves replacing the geometric structures that underlie General Relativity with non-commutative algebraic structures.

One way to approximate the classical continuous description of General Relativity with Quantum Mechanics is to introduce space-time deformation quantization. In this approach, points no longer exist and are replaced by Planck cells by inserting the Poisson tensor $\Theta^{\mu\nu}(x)$ into the standard theory. As a consequence, the commutator operation involving coordinate functions, in the leading order in the deformation parameter $\lambda$, reads [73,74]

$$[x^\mu, x^\nu] = i\lambda\Theta^{\mu\nu}(x) + \mathcal{O}(\lambda^2).\tag{49}$$

According to the authors of [73,74], coordinate uncertainty relations may have implications in the quantum micro-structure of space-time, smoothing out ultraviolet divergences, preventing gravitational collapse, and even allowing black hole formation only at scales larger than the Planck length.

Recently, in a non-commutative quantum mechanical approach based on the Seiberg–Witten map, the authors of [75] proposed a parametrization scheme that associates non-commutative parameters with the Planck length and the cosmological constant. As a result, they discovered that non-commutativity introduces an effective gauge field in the Schrödinger and Pauli equations, which breaks translation and rotational symmetries in the non-commutative phase-space, leading to the generation of intrinsic quantum fluctuation effects.

Moreover, in accordance with the uncertainty principle of Quantum Mechanics, quantum vacuum fields may possess an immense amount of energy. According to the equivalence principle of General Relativity, this energy must gravitate, producing significant gravitational effects. Consequently, fluctuations in the quantum vacuum, intrinsically associated with a non-commutative algebraic structure, could potentially serve as a cause for the expansion of the Universe. This aspect of cosmic acceleration served as the primary motivation for the study conducted based on the Hořava–Lifshitz theory and branch-cut gravitation. The study developed a formalism grounded in the non-commutativity of a "mini-superspace" of variables obeying Poisson's algebra.

Following a similar line of investigation, Massimiliano Rinaldi [76] proposes a novel scenario in which the inflationary phase is not driven by a classical scalar field, as introduced by Alan Guth [77,78], but rather by a non-commutative structure of space-time, whose dynamics are governed by quantum effects encoded in the expectation value of the stress tensor. Rinaldi introduces an inflationary scenario expressed in terms of coherent states of non-commutative quantum field theory [76]. The inspiration for this formulation is drawn from the work in [79], where it was demonstrated that the study of dispersion relations, involving a maximal momentum or, more precisely, a minimum Compton wavelength (quantum of space), could lead to inflation as the radiation temperature surpasses the Planck temperature. Rinaldi's formulation, in turn, is based on the concept that the non-commutative structure of space-time naturally regulates the divergent ultraviolet behavior of the propagator. Consequently, the stress tensor for matter fields should be finite in the UV domain. The results of this approach indicate the presence of a minimal length, which primarily contributes to smearing off the expectation value of the quantum stress tensor. Additionally, this formulation affects the acceleration behavior of the scale factor $a(t)$ of standard cosmology and cosmic density. A noteworthy point of comparison with branch-cut gravity pertains to the density of the Universe, which is no longer singular. As a result, it becomes possible to extend the analysis to include the pre-Big Bang scenario of string cosmology, allowing time to extend from $-\infty$ and $+\infty$ [76]. Furthermore, the analysis of the number of e-folds related to the asymptotic behavior of the scale factors $a(\mp\infty)$ yielded a value of $N = 60$ [76], which corresponds to $e^N = e^{60}$ e-folds. This result can be contrasted with the number of folding branches ($fb$) to achieve causality in BCG, which is estimated to be approximately $N_{BCG}^{[fb]} \sim 10^{61}$ [6].

Similarly to the findings presented in the work of Rinaldi [76], in the context of BCG non-commutative quantum cosmology, the inflationary phase is not driven by a classical scalar field [77,78]. Instead, it arises from the non-commutative structure of space-time, governed by quantum effects encoded in the Hamiltonian structure of the model. This quantum influence is explicitly manifested through the presence of a parametric non-commutative algebra.

The main outcomes of our approach suggest that the wave function of the Universe, as a function of the branch-cut scale factor, may exhibit, in addition to a wave-like behavior, systematically, amplitude decreasing in the contraction phase, and amplitude increasing during the expansion phases as well as a frequency increase. This intriguing behavior stems from the imposition of non-commutativity in a mini space-time superspace of variables obeying Poisson algebra. The implications of the non-commutative algebra can be visualized in the solutions of the wave function of the Universe, indicating in the expansion phase, a dynamical acceleration driven by a force whose work can be hypothetically synthesized in an expression of the type $W = -p_f dV$, where $p_f$ represents the strength of the pressure in the expanding region. This force may be originated, as the behavior of the potential $V(\eta)$ suggests, by the reconfiguration of matter in the early Universe due to the algebraic structure of the non-commutative geometry, by means of an asymmetric potential, that sets up the presence of primordial matter. The non-commutative algebraic structure captures the short-distance properties of space-time, with notable implications for the dynamical evolution of the branch-cut Universe. In short, our results indicate that Planck scale effects are encoded in the non-commutative group manifold structure, implying not

only an effective dimensional space-time reduction but also a reconfiguration of matter and fields, which in turn drives the acceleration of the Universe.

An interesting aspect concerns whether this type of inflation modeling may be completed in late times and then the reheating process can be carried out. The answer to this question currently lacks strong elements for its understanding. In the original version of the inflationary model, inflation is completed by a first-order phase transition, in which the Universe decays from its false vacuum state by bubble nucleation. In the first stage of reheating, vacuum energy is converted into kinetic energy for the bubble walls. Here, the expanding Universe is not created out of nothing, emerging instead from a previous stage, of contraction, with a transition region modeled by a smooth topological branch-cut structure of continuously connected Riemann surfaces with a new scale parameter, $\ln^{-1}[\beta(t)]$, analytically continued to the complex plane, the unique dynamical variable of the theory, stratified into hypersurfaces restricted to leaves of a Riemann foliation. A damping mechanism that causes this kind of inflation to be finalized and a later time reheating process to be realized represents an interesting challenge to be addressed in future studies.

**Author Contributions:** Conceptualization, C.A.Z.V.; methodology, C.A.Z.V., B.B., P.O.H., J.d.F.P., D.H., F.W., and M.M.; software, C.A.Z.V., B.B., M.R., and M.M.; validation, C.A.Z.V., B.B., D.H., P.O.H., J.d.F.P., and F.W.; formal analysis, C.A.Z.V., B.B., P.O.H., J.d.F.P., D.H., and F.W.; investigation, C.A.Z.V., B.B., P.O.H., J.d.F.P., M.R., G.A.D., M.M., and F.W.; resources, C.A.Z.V.; data curation, C.A.Z.V. and B.B.; writing—original draft preparation, C.A.Z.V.; writing—review and editing, C.A.Z.V., B.B., P.O.H., J.d.F.P., D.H., G.A.D., M.R., M.M., and F.W.; visualization, C.A.Z.V. and B.B.; supervision, C.A.Z.V.; project administration, C.A.Z.V.; funding acquisition (no funding acquisition). All authors have read and agreed to the published version of the manuscript.

**Funding:** P.O.H. acknowledges financial support from PAPIIT-DGAPA (IN100421). F.W. is supported by the U.S. National Science Foundation under Grant PHY-2012152.

**Data Availability Statement:** There is no data available.

**Conflicts of Interest:** There is no conflict of interest.

## Notes

1    $\ln^{-1}[\beta(t)]$ represents the inverse of $\ln[\beta(t)]$ and $\beta(t)$ represents a regularization function that identifies the range and cuts of the helix-like cosmological factor in branched gravitation [10].

2    $N(t)$ does not represent a dynamical quantity; instead, it denotes a pure gauge variable.

3    Bekenstein's criterion is understood here as a kind of criterion of truth [21], a measure of the truthfulness and reliability of our knowledge of the limits of validity of the model [7,22]

4    General Relativity is recovered in the limit $\lambda \to 1$, which corresponds to the full diffeomorphism invariance [19]

5    The canonical quantization Dirac procedure applied to the Einstein–Hilbert action results in a second-order functional differential equation defined in general terms in a configuration superspace, whose solutions depend on a three-dimensional metric and on matter fields [37–40]. Among the different quantization methods, the canonical quantization procedure allows the preservation of the original formal structure of a classical theory, as well as its symmetries and corresponding underlying conservation laws.

6    to simplify notation, the hat symbol is not used in the operators $\hat{u}$ and $\hat{p}_u$, or in most equations involving the time-dependent variable $\hat{u}$.

7    The coincidence problem refers to the initial conditions necessary to produce the quasi-coincidence of the densities of matter and quintessence in the current stage of the Universe [46,47].

8    Because the variables $u$ and $v$ commute, so do $\tilde{u}$, and $\tilde{v}$. Inspecting Equation (8), this implies that $\sigma$ is set to zero.

9    It is important to remember that the parameter $\chi$ that complements the proposed algebraic structure is implicitly inserted in the variables $\eta$ and $\xi$.

10    We draw attention to the fact that the difference in scales of the contraction and expansion phases of Figure 4 requires careful analysis to confirm these statements.

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
