# Peer review of "A Wheeler–DeWitt Non-Commutative Quantum Approach to the Branch-Cut Gravity"

_universe, doi:10.3390/universe9100428_

Round 1
Reviewer 1 Report
In this work, the authors have discussed non-commutative algebra and its consequences in describing a quantum universe utilizing the theory of Hoˇrava-Lifshitz gravity. It is an interesting issue and can be published in the journal. Many works have been done in the field of non-commutative algebra, and for different gravities, among them, I can refer to work 1506.08416 [pdf, ps, other] which has investigated different vacuua in a mini super space sub-algebra.
Author Response
Lifshitz

Reviewer 2 Report
In this paper, the authors analyze the branch-cut gravitation (BCG) in Horava-Lifshitz theory of gravity. The authors have conducted a study of Wheeler-DeWitt equation in Horava-Lifshitz gravity with BCG. The results are interesting and its results deserve to be published.
Before I have some comments on the paper and some points deserve to be clarified.
1. In action (2) the authors did not define the N and \lambda and variables. In addition there are some typos in this equation, there are missing subindices in letter M (Planck mass).
2. In equation (3) the authors did not define the \beta function.
3. In equation (9) explain the what kind of transformation is considering. Is it a canonical transformation? Otherwise is it not quite arbitrary?
4. Also on the issue of the uncertainty relation (46). In the equation (46) if one takes the limit \theta \to 0, the limit seems to be singular. Is it so? Please briefly discuss the remaining theory in this limit.
Reviewer 3 Report
In this paper, the effects of non-commutativity of a mini-superspace of variables with the Poisson algebra are studied on the structure of the branch-cut scale factor and on the acceleration of the universe. The manuscript is quite well written and the mathematical results are given in detail. If the following points are taken into account, this work could be reconsidered for publication.
1) There are the past related works on the non-commutativity from string theories and the quantum cosmology with a Wheeler-DeWitt equation in the literature. In comparison with the preceding works, it is recommended that the new points and important progress found in this work should be explained in more detail and explicitly.
2) What are the fundametal physical reasons (mechanisms) why the non-commutativity leading to a modified branch-cut gravitation with a quantum scale factor can yield the cosmic acceleration, i.e., inflation?
3) Related to the point 2), can this kind of inflation be finalized and then can the reheating process be realized?
Minor editing of English language required
Round 2
Reviewer 2 Report
The authors have answered satisfactorily all the quastions and comments. I am happy to accept the article for publication.
Reviewer 3 Report
The authors' answers and revision are appreciated. In the revised manuscript the points in the report are taken. Therefore the revised manuscript could be suitable for publication.
Minor editing of English language required